# Comprehensive metabolomics of Philippine *Stichopus cf. horrens* reveals diverse classes of valuable small molecules for biomedical applications

**Vicenzo Paolo M. Torreno[1], Ralph John Emerson J. Molino[2], Hiyas A. Junio[2], Eizadora T. Yu[1]***

**1** The Marine Science Institute, University of the Philippines, Diliman, Quezon City, Philippines, **2** Institute of Chemistry, University of the Philippines, Diliman, Quezon City, Philippines

* etyu@up.edu.ph

**Data Availability Statement:** The authors confirm that the data supporting the findings of this study are available within the article and its supplementary materials. In addition, the compiled

## Abstract

*Stichopus cf. horrens* is an economically important sea cucumber species in Southeast Asia due to their presumed nutritional and medicinal benefits. However, compared to other sea cucumbers such as *Apostichopus japonicus*, there are no biochemical studies on which compounds contribute to the purported bioactivities of *S. cf. horrens*. To address this, a high-throughput characterization of the global metabolite profile of the species was performed through LC-MS/MS experiments and utilizing open-access platforms such as GNPS, XCMS, and metaboAnalyst. Bioinformatics-based molecular networking and chemometrics revealed the abundance of phospholipids such as phosphatidylcholines (PCs), phosphatidylethanolamines (PEs), phosphatidylinositols (PIs), and phosphatidylserines (PSs) in the crude samples. Body wall extracts were observed to have higher levels of structural, diacylated PCs, while the viscera have higher relative abundance of single-tail PCs and PEs that could be involved in digestion via nutrient absorption and transport for sea cucumbers. PEs and sphingolipids could also be implicated in the ecological response and morphological transformations of *S. cf. horrens* in the presence of predatory and other environmental stress. Interestingly, terpenoid glycosides and saponins with reported anti-cancer benefits were significantly localized in the body wall. The sulfated alkanes and sterols present in *S. cf. horrens* bear similarity to known kairomones and other signaling molecules. All in all, the results provide a baseline metabolomic profile of *S. cf. horrens* that may further be used for comparative and exploratory studies and suggest the untapped potential of *S. cf. horrens* as a source of bioactive molecules.

## Introduction

The Philippines is home to diverse marine life, including sea cucumbers that inhabit coastal areas throughout the vast archipelago [1]. Among 170 reported species, 41 are commonly

MS data files were uploaded to the CCMS (Center for Computational Mass Spectrometry, UCSD) repository. Reviewers can access the dataset through the link: https://massive.ucsd.edu/ProteoSAFe/static/massive.jsp. The dataset can be specifically searched by going to the MassIVE Datasets Tab and typing MSV000090651 in the search box. (DOI: https://doi.org/doi:10.25345/C5WM13Z5C).

**Funding:** ETY received funding for the study from the Department of Science and Technology–Philippine Council for Agriculture, Aquatic, and Natural Resources Research and Development. The funders had no role in study design, data collection, analysis, the decision to publish, or the preparation of the manuscript.

**Competing interests:** The authors have declared that no competing interests exist.

foraged by fisherfolk, and considered to be economically important [1]. The huge commercial demand for *Cucumaria frondosa*, *Holothuria scabra*, *Apostichopus japonicus* is driven by the abundance of valuable molecular constituents such as saponins and fucoidans in these species [2–5]. In the Philippines, *Stichopus cf. horrens* (Fig 1) is listed as one of high-value species involved in the trepang trade [1, 6, 7]. In some countries such as Malaysia, they are sold as "gamat" in dried and processed forms due to their medicinal and nutritional properties [8]. Unfortunately, extensive biochemical investigations are lacking for this species, which based on its ecological adaptation, is a promising source of bioactive compounds. Known to the locals as "hanginan" [exposed to air], S. horrens tends to quickly shed off its body wall in response to touch stimulus [9, 10]. Gamat oils and gels have been used as effective traditional medicines for wound healing, and this has been substantiated by animal studies with crude sea cucumber extracts [11–13]. Identification of sea cucumber metabolomes, therefore, can reveal the vast molecular diversity found in these animals, and provide insights on compounds with potential bioactivities and nutritional benefits that can be targeted for future studies [5, 14]. In addition, curation of metabolomes from different organisms can assist in determining sustainable sources of bioactive compounds, given that similar compounds can be found in several organisms, albeit at different quantities.

Metabolomics could contribute valuable information through the detection, characterization, and comparative analysis of high-value and biologically relevant small molecules from marine echinoderms [14]. The usefulness of metabolomics has been especially demonstrated in the study of *A. japonicus*. Interspecies variation and geographic discrimination were demonstrated through metabolite fingerprinting and multivariate statistics [5, 15]. Pathway analysis also revealed differentially expressed metabolites during the evisceration process [16]; in the presence of thermal stress and hypoxia [17]; and skin ulceration syndrome [18]. Literature on specific biological aspects of *A. japonicus*, however, did not include comprehensive characterization of small molecules outside statistically significant features corresponding to primary metabolites (amino acids and derivatives) and a number of lipids [5, 15–18].

Previous studies on *S. cf. horrens* focused mainly on bioassay-guided isolation of cytotoxic saponins such as Stichoposides A-D and Stichorrenosides A-E [19, 20]. More recently, rapid profiling of sea cucumber saponins using on-line Ultra-Performance Liquid Chromatography

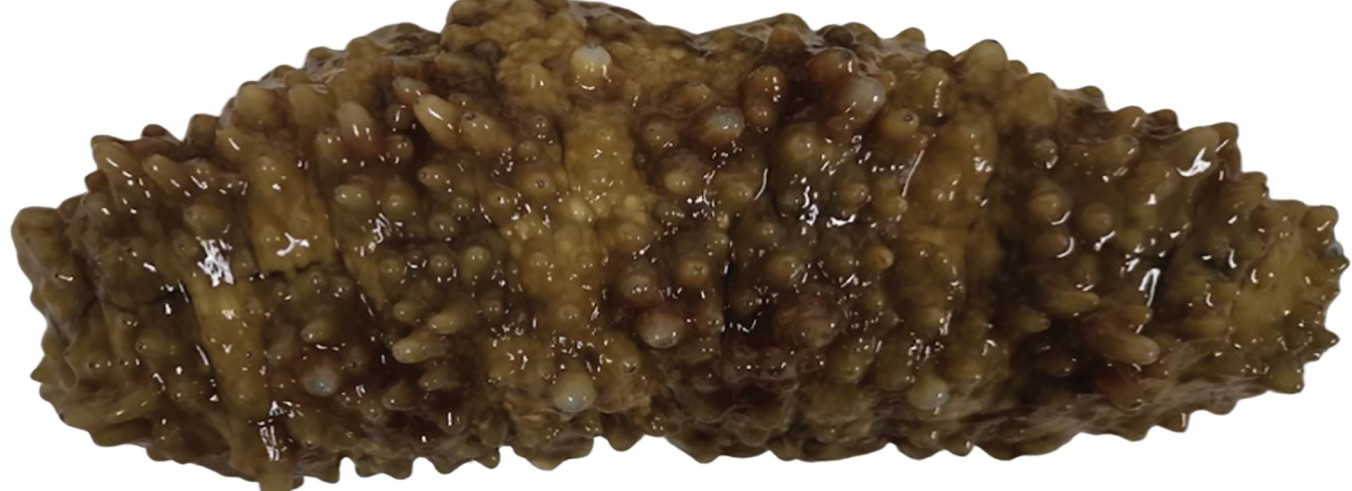

**Fig 1. Adult *Stichopus c.f. horrens* harvested from Anda, Pangasinan.** The dermis consists of characteristic spicules that protect the softer body. (©Vicenzo Torreno).

Quadrupole Time-of-Flight (UPLC®-QTOF) mass spectrometry and manual fragmentation analysis was demonstrated [19]. This work is expanded herein towards identifying other classes of compounds that can be found in the dermis and viscera of *S. cf. horrens*. Sea cucumber body wall is extensively traded for culinary purposes and as traditional medicines [1]. Interestingly, the viscera are collected and sold as tonic in some communities but is typically discarded during the processing of beche-de-mer. In-depth structural and comparative analysis must be performed as a necessary groundwork for the exploration of the molecular components of sea cucumbers.

Characterization of the sea cucumber compounds in this work makes use of community-curated data in cloud-based Global Natural Products Social Molecular Networking (GNPS) [21]. Molecular networking analysis was employed to cluster structurally related compounds based on similar fragmentation patterns [21]. Moreover, MolNetEnhancer, which is an integrated workflow with advanced computational tools such as MS2LDA, was used to detect and classify spectral motifs to further improve annotation of spectral families [22, 23]. Considering these recent advances in metabolomics data analysis, detailed herein is a comprehensive workflow for the characterization and comparative analysis of the global metabolite profile of sea cucumbers. This study will complement further ecological investigations, systems biology studies, and research & development (R&D) of sea cucumber-derived products, and potentially contribute to the blue economy initiatives of tropical coastal communities.

## Materials and methods

### Chemical reagents

HPLC grade absolute ethanol utilized in the extraction of small molecules was from Scharlab (Barcelona, Spain). LC-MS grade methanol and water used in the LC-MS analysis were from Merck (Darmstadt, Germany). Concentrated formic acid was supplied by Thermo Fisher Scientific (Massachusetts, USA). Whatman filter papers (Grades 1 and 3) were from Cytiva (Massachusetts, USA).

### Sampling and extraction of secondary metabolites from *S. cf. horrens*

Ten *S. cf. horrens* adult individuals weighing around 1kg were obtained from cultures at the Bolinao Marine Laboratory, Marine Science Institute at Bolinao, Pangasinan. Samples were killed by freezing and transported with ice to the Institute of Chemistry, University of the Philippines, Diliman. Samples were immediately stored in -80˚C prior to sample processing.

The whole body wall and viscera were separated and minced. Fifty grams of the minced body wall for each animal was extracted with absolute ethanol (1:20 g/mL) for 48 hrs. The viscera were extracted as a whole at the same ratio. Thereafter, the samples were filtered and dried in vacuo using a rotary evaporator (Stuart RE300). Dried extracts were then resuspended with absolute ethanol to a final concentration of 10 mg/mL for LC-MS analysis.

### UPLC-QTOF analysis

LC-MS analysis were carried out in a Waters Acquity®—UPLC system in tandem with a Waters Xevo G2-XS quadrupole time-of-flight mass spectrometer. Chromatographic separation of extracts from S. horrens utilized a Waters Acquity® BEH Fluoro-Phenyl stationary phase (1.7 μm, 50 mm x 2.1 mm) and binary mobile phase of acetonitrile: water infused with 0.1% formic acid. Optimum separation was obtained using gradient elution with %acetonitrile as follows: 5% until 0.75 mins, 40% at 1.00 min, 65% at 5.00 mins., 80% at 9.00 mins., 100% at 9.50 mins, and back again to 5% at 10.00–10.50 mins.

The initial profiling was carried out in the positive mode by specifying a capillary voltage of 2.80 kV, cone voltage of 40.00 V, offset potential of 80.00 V, and source and desolvation temperatures of 150°C and 500°C, respectively. Negative mode analysis was carried out under the following parameters: capillary voltage: 1.80 kV, cone voltage: 40.00 V, offset potential: 80.00 V, source temperature: 150°C, desolvation temperature: 500°C. For both positive and negative analyses, acquisition of mass spectra was through a 0.50 second scan time within m/z 50–2000 mass range. The same scan time and precursor window was used in the Data-Dependent Acquisition (DDA) of MS/MS spectra of the highly intense features in the sample. Criteria for the selection of precursor ions include an intensity threshold of $3.0 \times 10^5$ ion counts and the top 6 most intense product ions per scan time.

To obtain extensive information on the fragmentation patterns, collision-induced dissociation (CID) using argon was separately carried out utilizing fixed and ramped collision energies (15 eV, 15–30 eV, 30 eV, 30–45 eV, 45 eV, and 45–60 eV for compounds less than 1kDa and 60 eV, 75 eV, and 90 eV otherwise).

## Bioinformatics analysis

Waters.RAW files were converted to open source, mzXML format, using the MSConvert module of Proteowizard. Files were then uploaded to the Center for Computational Mass Spectrometry (CCMS) repository of the University of California San Diego (UCSD) before analysis on GNPS.

Putative Identifications for the compounds were provided through Library Search in the public spectral libraries of GNPS. Parameters for spectral matching include precursor and product ion mass tolerances of m/z 0.02 and m/z 0.05, respectively. Spectral matches considered have at least eight matched peaks and a calculated cosine (similarity) score greater than 0.70.

In addition, structural relationships among precursor ions were presented in terms of molecular networks that were also generated by GNPS. Data was pre-processed using MZmine 2.52 to deisotope and deconvolute precursor ions [24]. Set of parameters used are summarized in S9 Table. After the gap-filling step, necessary processes for the generation of ion-identity molecular networks were executed. Ion correlation required feature shape and height correlation (Pearsons) of 85% and 50%, respectively. The.csv and.mgf files were then exported and uploaded to the feature based molecular networking (FBMN) module of GNPS. As suggested for high-resolution instruments, the precursor and product ion mass tolerance used for the clustering of identical spectra were m/z 0.02 and m/z 0.05, respectively. Individual nodes, representing unique precursor ions, were connected if the similarity (cosine) score between them exceeded 0.70. Visualization of molecular networks was done through Cytoscape 3.8.2 [25]. Ion-identity molecular networks in the expanded forms are presented and discussed in this study. To present associated features detected in either positive and negative mode, merged polarity networks were also generated using a specialized workflow in GNPS that require individual positive and negative ion mode networks as input.

Further information on shared structural backbones was obtained through MS2LDA and MolNetEnhancer analysis on GNPS. Identification of fragmentation motifs involved assessment of overlap score and probability threshold values that were both set to be above 0.30. In addition, MolNetEnhancer readily integrated the results from classical molecular networking and MS2LDA without requiring predisposed criteria and parameters.

## Quantification of sulfated compounds

Determination of the amount of sulfated compounds was through a modification on the method originally published by Hayashi (1975) [26]. Alkyl sulfates are considered methylene

blue active substances (MBAS) that form a 1:1 complex with methylene blue. Reactions were carried out by combining 110 uL methylene blue (0.1%, pH 7.2 phosphoric acid buffer) with 220 uL = sample solutions (10 mg/mL). Methylene blue chromophores that interacted with sulfated surfactants were then extracted with two washings of 660 uL ethyl acetate. Organic solvent was evaporated, and the dried residue was resuspended with 150 uL DMSO. The absorbance of the solution at 665 nm was recorded using a ThermoFisher Multiskan GO spectrophotometer. The relationship between absorbance and concentration of MBAS were obtained through a calibration curve (2.5–40 ppm) using sodium dodecyl sulfate (SDS) standard.

## Results

### UPLC®-QTOF profiling of *S. cf. horrens* extracts

Metabolomics of *S. cf. horrens* was carried out to detect, annotate, and compare small molecules present in the crude extracts. In order to maximize the coverage of the annotated metabolites, full-scan experiments were performed in both positive and negative ionization modes. In addition, data-dependent acquisition of tandem-MS spectra were carried out using different collision energies to obtain comprehensive information on the fragmentation patterns of different compounds. Representative chromatograms from the analyses of the body wall and viscera are shown in S1 Fig. Various analyses in the GNPS pipeline were executed to annotate the metabolites corresponding to highly intense signals. Examples of tail-to-tail spectral matches are presented for a polyunsaturated fatty acid (PUFA) (Fig 2A), phosphocholine (Fig 2B), phosphoethanolamine (Fig 2C), and UDP-GlcNAc (Fig 2D), a key metabolite involved in the biosynthesis of neuraminic acid lipids [27]. Specific classes of compounds eluted in distinct regions of the chromatogram. Saponins were observed between $t_R$ = 2.50 to 3.25 mins. Detected only in the positive mode are acylcarnitines ($t_R$ = 2.72 to 3.01 mins.), which were immediately followed by single-tail phosphatidylcholines and phosphatidylethanolamines ($t_R$ = 3.52 to 5.58 mins.) Also, within the 5-minute mark, the presence of polyunsaturated fatty acids was noted. This was preceded by phosphatidylinositols ($t_R$ = 6.17 to 9.33 mins.), and phosphatidylserines ($t_R$ = 7.23 to 8.82 mins.) Also with a broad range of elution profiles are double-tail phospholipids ($t_R$ = 5.47 to 10.23 mins.) and sphingolipids ($t_R$ = 5.66 to 10.05 mins.).

Comprehensive details of metabolite annotation are tabulated in S1–S8 Tables with a summary presented in Fig 3A. The most widespread compounds found in both the body wall and internal organs are phosphatidylcholines, followed by sphingolipids and phosphoethanolamines. Sulfated alkanes and sulfated sterols, phosphoinositols, as well as few holostane-type triterpenoids and their glycosylated and sulfated derivatives also appear to be abundant in the samples. Meanwhile, taurines appeared to be exclusively found in the body wall while acylcarnitines were localized in the viscera. The solvent used in partitioning appears to influence the kinds of compounds that were extracted and detected by MS (Fig 3A and S1–S8 Tables), for example, with isobutanol partitioning favoring the amphipathic saponins and phosphocholines. In contrast, signal intensities of sphingolipids appear to decrease with solvent-partitioning, while the levels of sulfated compounds remain unaffected.

Aside from compound identification, peak areas obtained from data pre-processing were grouped together and compared to present a summary of detected signals in the positive (Fig 3B) and negative mode (Fig 3D). Observed higher ionization efficiency of the choline head group in the positive mode compared with the negative mode can account for the 71.1% versus. 6.90% peak area contribution, respectively, of phosphocholine. Meanwhile, phosphoethanolamines (35.2%), sulfated compounds (15.5%), and high molecular weight saponins (9.2%)

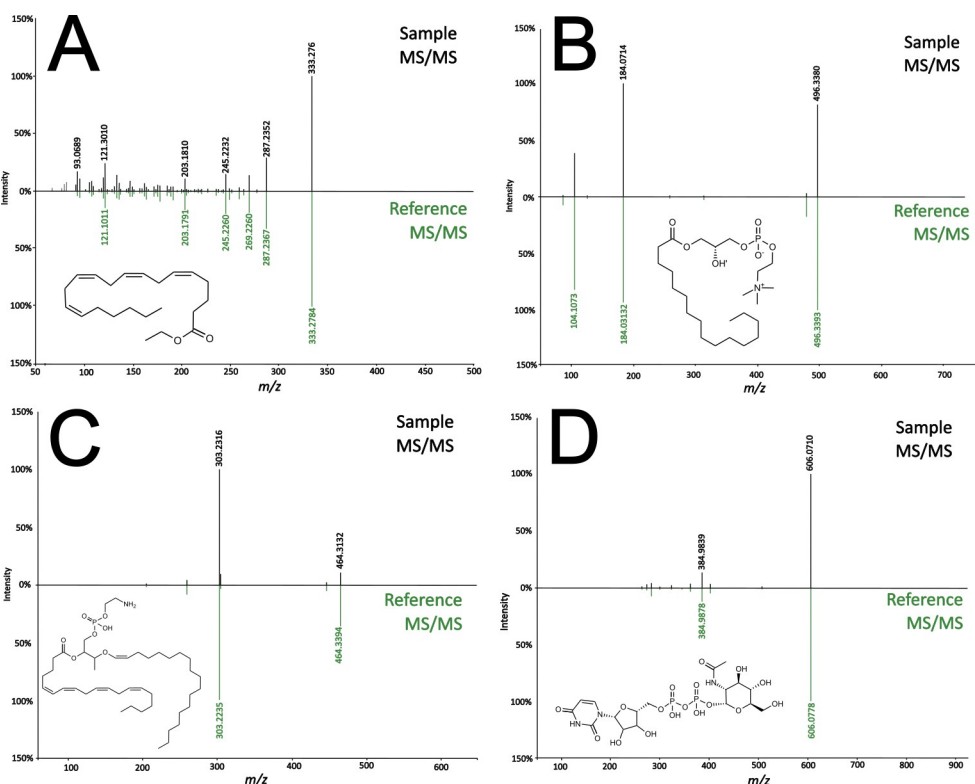

**Fig 2. Tail-to-tail matches leading to the annotation of high-value small molecules from *S. cf. horrens*.** Spectral matches are shown for (A) arachidonic acid ethyl ester (cosine = 0.83, 3.27 ppm), (B) PC(16:0) (cosine = 0.95, 0.26 ppm), (C) PE(P-18:0/18:1) (cosine = 0.76, 3.13 ppm), and (D) Uridine-5-Diphospho-N-acetylglucosamine (cosine = 0.94, 2.23 ppm).

were preferentially ionized in the negative mode due to the presence of acidic functional groups or the favorability of forming the anionic [M+HCOO]⁻ adduct [28]. Outside these compounds, 31.2% of the detected signals in the negative mode were not linked to any annotation, which is more than double of unclassified features (14.7%) in the positive mode.

After accounting for the individual structures and signal contribution of the different classes of compounds, structural diversity within the group of small molecules produced by S. horrens was examined. A python script based on the rdkit package was written to calculate the average cosine scores of molecules from their SMILES strings (S1 File). Major finding was a close structural semblance (cosine > 0.80) attributed to the terpenoid backbones of saponins and polar head groups of lipids. Molecular species differ only in carbon chain length; positions of double bonds; and presence of additional glycosides and other residues (such as acyl and sulfate groups for saponins) (Fig 3B and 3E). Structural relatedness suggests shared biochemical and physiological roles of a compound class [29–31].

## Molecular networking and motif analysis

Significant structural semblance observed in tandem-MS spectra were further investigated and projected in the molecular network in Fig 4. As expected, related precursor ions clustered into spectral families as their cosine scores exceeded the 0.70 threshold set [21]. Presentation of metabolites in Fig 4 also utilized more advanced analyses in GNPS such as feature-based molecular networking to distinguish isobaric features, as well as ion-identity molecular

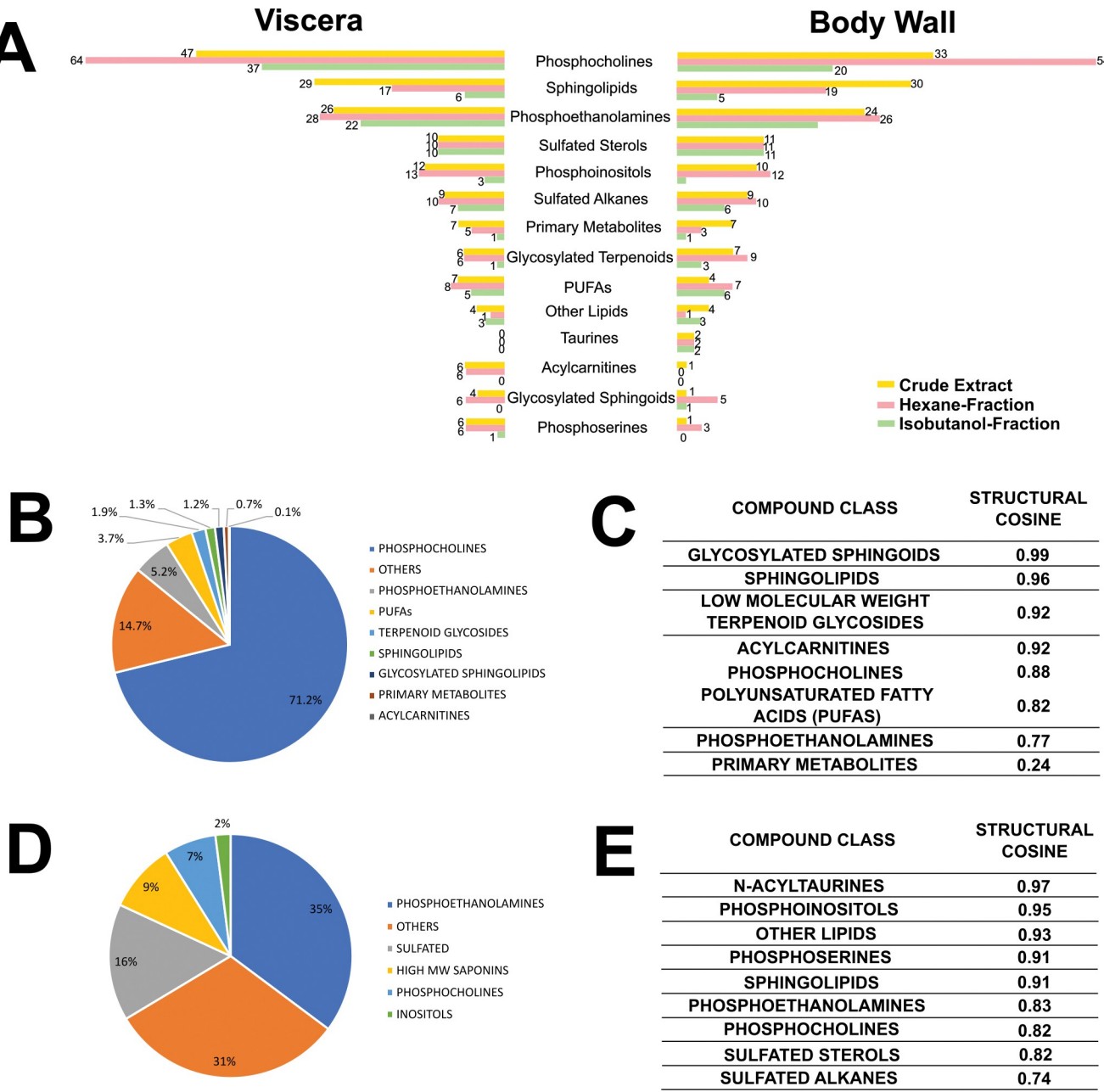

**Fig 3. Summary of compound annotation for the different extracts prepared from *S. horrens*.** The presence of different classes of small molecules between the body wall and viscera are presented in (A) in different solvent fractions: crude ethanolic extracts (yellow), isobutanol fractions (green) and hexane fractions (red). Also presented are the percentages of signals contributed by each group of compounds in the positive (B) and negative (D) mode. Furthermore, information about within-group structural similarity are indicated in tables (C) and (E), respectively.

networking to group together and collapse into a single nodal representation, different ion forms (adducts) of the same compound [21, 32]. For ease in viewing, merged polarity networking was also carried out to combine positive and negative mode spectral families [21]. A summary of metabolites which were found both in the positive and the negative mods are summarized in S10 Table. Overlap is highlighted in Fig 4 through the connections between blue (positive) and red (negative) nodes.

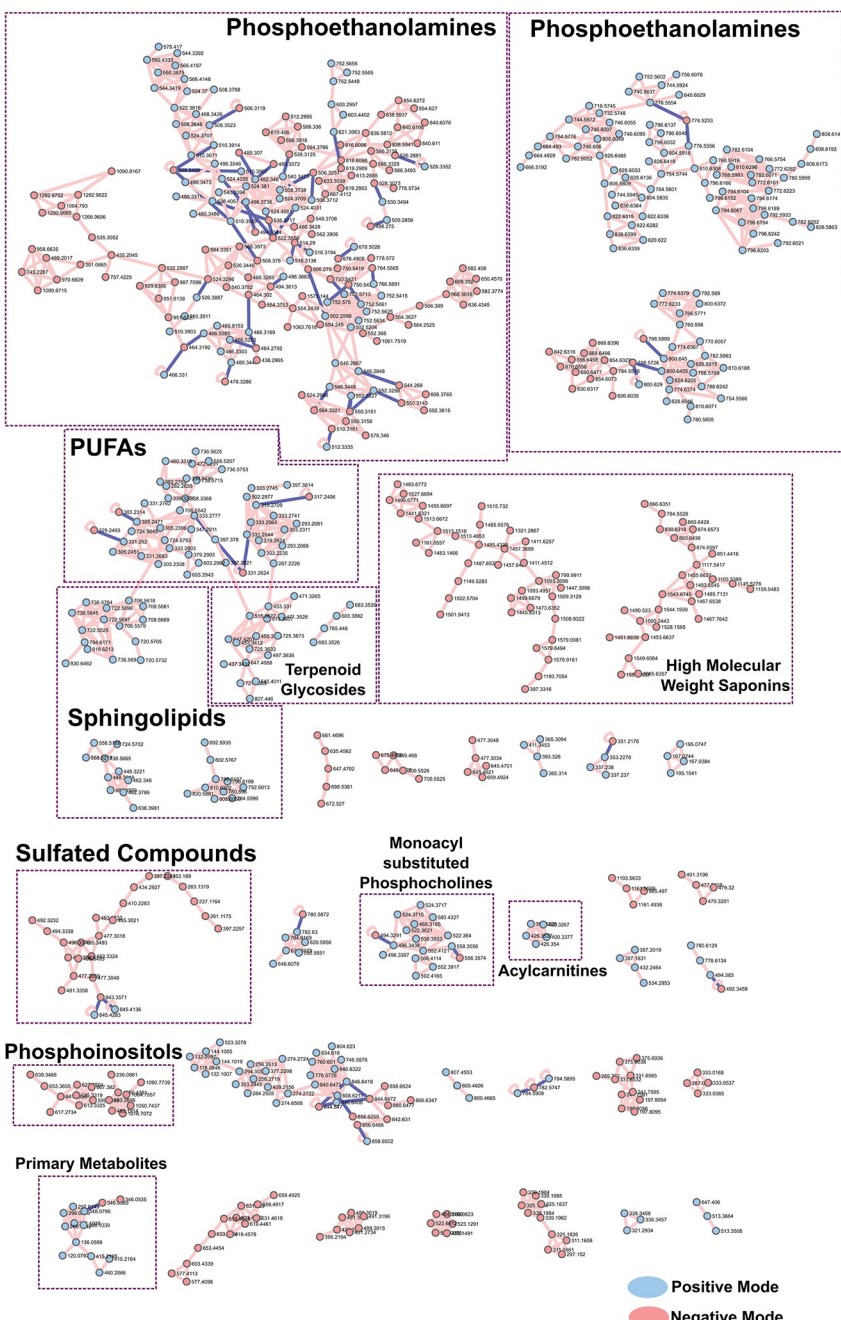

**Fig 4. Molecular network showing diverse classes of biomolecules from *S. cf. horrens*.** Nodes colored in blue and red correspond to spectra acquired from positive and negative mode DDA analysis, respectively. Self-looping nodes were omitted for simplicity.

Aside from overall spectral similarity, repeated patterns across spectra were identified by MS2LDA as individual Mass2Motifs [23]. For both the positive (Fig 5A) and negative (Fig 5B) modes, the majority of the motifs (nodes) were not linked to any existing descriptions (orange). To address this, suggested annotations are tabulated in Table 1, which were all based on known diagnostic ions and theoretical fragmentation mechanisms 23, 28, 29) for amphiphilic compounds in *S. horrens* (Fig 5). As expected, major structural motifs in the positive

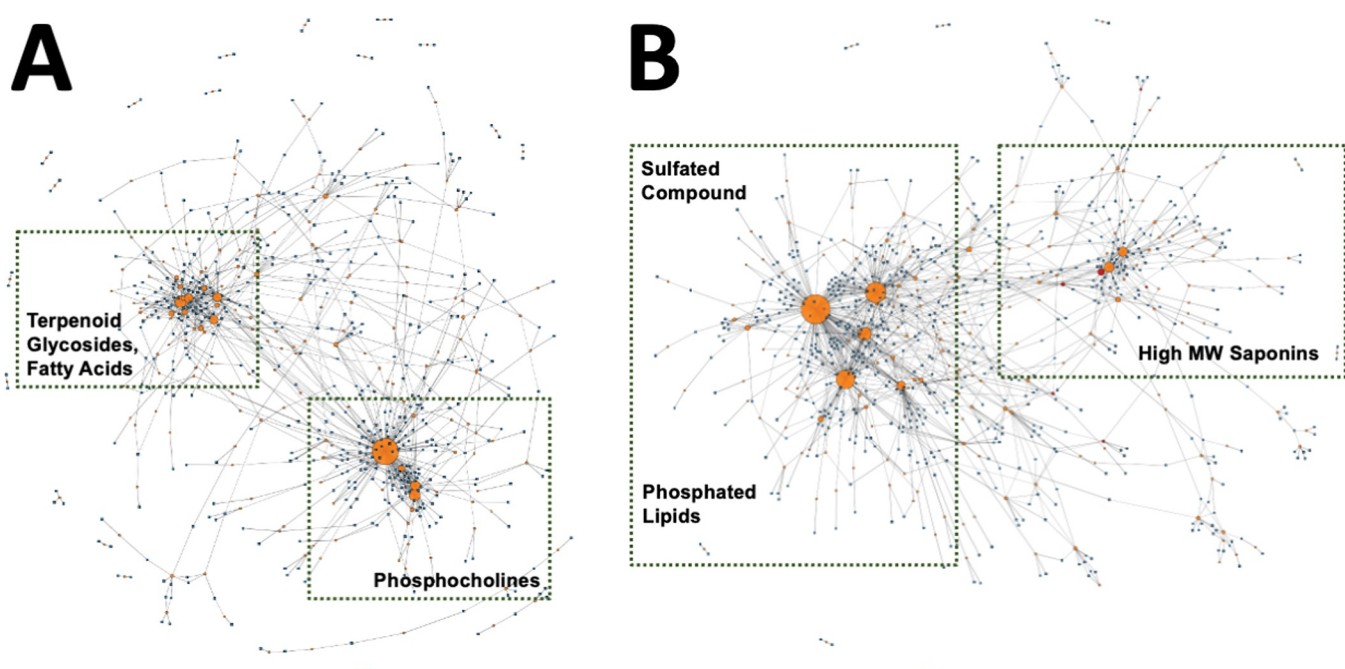

**Fig 5. Visualization of spectral motifs analyzed by MS2LDA.** Mass2Motifs are shown in orange, with node size proportional to the number of spectra (small rectangles) containing these motifs. Maps of spectral motifs from positive and negative mode are shown in (A) and (B), respectively.

mode were associated with phosphocholines identified in the samples. Motifs 16, 73, 250, and 270 were all derived from remote H-rearrangement leading to the formation of product ions coming from the phosphocholine head group ([29], S2A Fig). Similar rearrangement reactions also led to the fragmentation of phosphocholines, that existed as [M+HCOO]$^-$ adduct in the negative mode [30, 31]. Neutral loss of 60 Da was common among the ions in the spectra and can be attributed to the nucleophilic attack of the formate anion with a methyl group of the quaternary amine [33, 34]. In addition to this, the presence of phosphocholine head group was further verified by weakly intense diagnostic ion with m/z 168.044 (motif_2) [35]. Meanwhile, cleavage of the ester bonds liberated free fatty acyl groups, the mass-to-charge ratios of which were used to discern the chain length and degree of unsaturation of the hydrophobic tails of these lipids [35].

Common to all phospholipids is the m/z 152.996 product ion from dehydration and penta-cyclic ring formation within a phosphoglycerol anion [36]. Considering this, highly intense phosphoethanolamine lipids can be further distinguished based on the unique occurrence of m/z 196.031 (motif_134) coming from dehydrated glycerophosphoethanolamine [36]. Similarly, acylcarnitines, exclusively detected in the positive mode, can be readily annotated through the presence of m/z 85.028 formed from charge migration reactions from the carnitine head group [37]. In contrast, sphingolipids were characterized based on the loss of hexosyl group (-162 Da and -180 Da), as well as product ions coming from the protonated sphingoid backbone (motif_209, motif_290) [38].

Different from sphingolipids, the hexose group is included in the negative mode base peak ion (m/z 241.012, part of motif_255) of phosphatidylinositols predictably because the sugar residue is affixed with a phosphate group [36]. Also detected exclusively in the negative mode are phosphoserines characterized by a smaller neutral loss of 87.06 Da (motif_620) coming from the alanine ($C_2H_5NO_2$) esterified to phosphate [36].

**Table 1. Mass2Motifs derived from MS2LDA annotated through theoretical fragmentation of experimental tandem-MS.**

| | Ionization Mode | No. of Spectra | Motif Name | Main Product Ion/s and Neutral Loss/es | Annotation |
|---|---|---|---|---|---|
| 1 | POSITIVE | 132 | motif_250 | *m/z* 184.08 | Phosphatidylcholines |
| 2 | | 50 | motif_73 | *m/z* 86.09 | Phosphatidylcholines |
| 3 | | 41 | motif_16 | *m/z* 104.11 | Phosphatidylcholines |
| 4 | | 26 | motif_270 | *m/z* 124.99 | Phosphatidylcholines |
| 5 | | 15 | motif_39 | *m/z* 85.03 | Acylcarnitines |
| 6 | | 14 | motif_209 | *m/z* 250.26, *m/z* 238.26, *m/z* 250.26, *m/z* 264.28, *m/z* 268,27, -180Da | Sphingolipids |
| 7 | | 13 | motif_125 | *m/z* 77.04, *m/z* 119.06, *m/z* 93.05, -118Da, -135Da, -93Da | Fatty Acids, PUFAs |
| 8 | | 7 | motif_290 | *m/z* 260.24, -180Da | Sphingolipids |
| 9 | | 7 | motif_91 | *m/z* 184.08, *m/z* 796.62 | Phosphatidylcholines, [2M+Na]+ |
| 10 | | 6 | motif_101 | *m/z* 159.12, *m/z* 119.09, *m/z* 173.14, *m/z* 161.14, *m/z* 133.11, *m/z* 107.09, *m/z* 147.12, *m/z* 281.23 | Triterpenoid Glycosides |
| 11 | | 6 | motif_0 | *m/z* 282.28, *m/z* 264.27 | Sphingolipids |
| 12 | | 5 | motif_14 | *m/z* 262.26, -162.13 Da, -180.14 Da | Sphingolipids |
| 13 | | 5 | motif_99 | *m/z* 264.28 | Sphingolipids |
| 14 | | 5 | motif_224 | *m/z* 274.26, *m/z* 257.23, -162.13Da, -180.14Da | Sphingolipids |
| 15 | NEGATIVE | 159 | motif_454 | *m/z* 96.96 | Sulfated Compound |
| 16 | | 100 | motif_460 | *m/z* 79.96 | Sulfated Compound |
| 17 | | 40 | motif_518 | *m/z* 233.09, *m/z* 96.95 | Sulfated Compound |
| 18 | | 2 | motif_396 | m/z 167.03, -60.04 Da | Phosphatidylcholines, [M+HCOO]- |
| 19 | | 11 | motif_395 | -60.04 Da | Phosphatidylcholines, [M+HCOO]- |
| 20 | | 109 | motif_521 | -78.99 Da | Phosphated Compound |
| 21 | | 13 | motif_620 | *m/z* 152.99, -87.06 Da | Phosphatidylserines |
| 22 | | 14 | motif_255 | *m/z* 241.02, *m/z* 315.05 | Phosphatidylinositols |
| 23 | | 8 | motif_267 | *m/z* 152.99 | Phospholipids |
| 24 | | 20 | motif_539 | *m/z* 235.10 | Unidentified, prevalent |
| 25 | | 30 | motif_633 | *m/z* 303.23, *m/z* 259.24, *m/z* 78.96, *m/z* 96.97, -59.93 Da | Phosphatidylcholines, [M+HCOO]- |
| 26 | | 11 | motif_134 | *m/z* 196.04 | Phosphatidylethanolamines |
| 27 | | 3 | motif_163 | -46.08 Da | HCOOH adduct |

Outside phospholipids, the majority of the spectral features in the negative mode show highly intense m/z 96.960 product ion (motif_454), which could be an $HSO_4^-$ or $H2PO_4^-$ anion [39]. Annotations of sulfated compounds were further supported by m/z 79.957 (motif_460) coming from homolytic cleavage of the sulfodiester bond [40].

## Structural characterization of saponins

Compared to limited information about sulfated compounds in the negative mode, sulfated terpenoid glycosides were annotated with more confidence in the positive mode based on very extensive fragmentation detail. Characterization of the unreported metabolite in Fig 6A & 6B was based on losses of sulfur trioxide ($SO_3$, -80Da), acetyl group ($CH_3COOH$, -60 Da), as well as xylopyranoside residue ($C_5H_8O_4$, -132 Da) [39–42]. In addition to this, reactions such as retro-Diels Alder, decarboxylation, and decarbonylation were used to support the occurrence of a Holost-7-ene-3,23-diol parent structure (Fig 6B) [42]. This backbone is shared by sticho-poside and stichloroside saponins identified from our *S. cf. horrens* crude extract. The same fragmentation principle was used to account for the product ions of stichorrenoside A (with

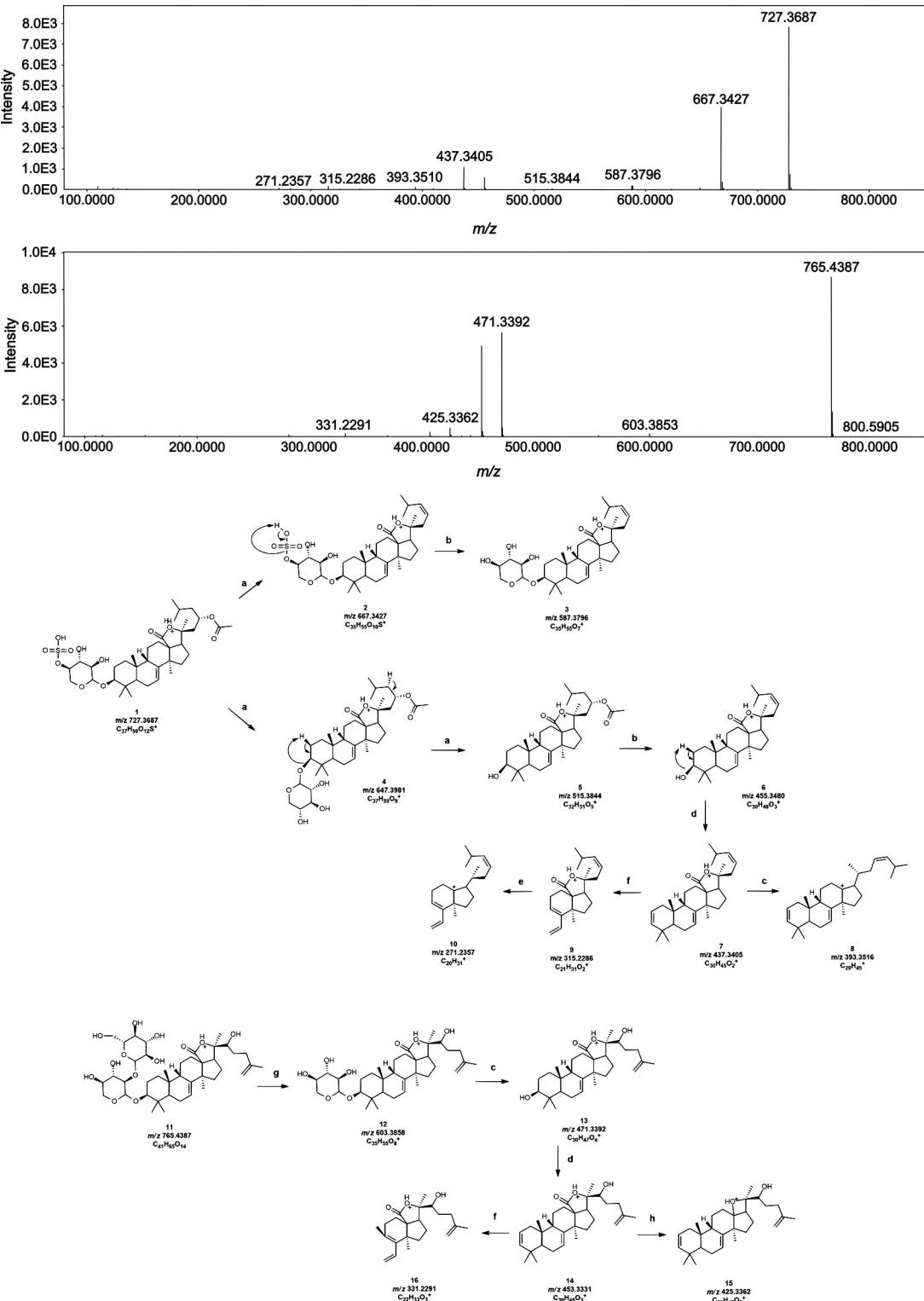

**Fig 6. Structural annotation of an unreported sulfated terpenoid glycoside and the known Stichorrenoside A from *S. cf. horrens*.** Tandem MS spectra for the mentioned compounds are shown in (A) and (B), respectively. For the reactions (a) loss of $SO_3$ (-80 Da), (b) elimination of $CH_3COOH$ (-60 Da), (c) deduction of xylopyranoside (-132 Da), (f) decarboxylation (-44 Da), (g) loss of glucoside (-162 Da), and (h) decarbonylation (-28 Da).

precursor m/z of 765.439). This compound readily lost glucoside (-162 Da) and xylopyranoside (-132 Da) sugar residues, consecutively ([37], Fig 6D). The proposed mechanisms readily accounted for the fragment at m/z 331.229 through the pericyclic ring fission (Fig 6D) of a Holosta-7,25-diene-3,23-diol scaffold.

We also observed that most lower MW saponins that appear to lack one or more sugar residues always co-eluted with high MW saponins, rather than eluting independently as the other saponins. Thus, these lower MW saponins were presumed to be products of in-source fragmentation. Other studies on oligosaccharide fatty acid esters also show in-source fragmentation of the sugar chain at different cone voltages [43]. Saturated and unsaturated pairs of annotated saponins (such as m/z 1475 and 1477, and m/z 1461 and 1463) are consistently accompanied by m/z 807, 645, 513 for the annotated, unsaturated saponins and m/z 809, 647, 515 for the saturated saponins. MS2 analysis of these in-source fragments also shows similar patterns suggesting that the aglycone of these ions also have similar structures. For instance, MS2 peaks generated from the m/z 809 ion were found to contain m/z 647 and m/z 515. Moreover, the MS2 peaks for these ions, especially below m/z 500 are extremely similar.

The MS2 spectra of the m/z 809 ion matches the theoretical fragmentation of Stichoposide B shown in S3 Fig, whose structure is very similar to Stichloroside B1 but with only two (2) sugar residues instead of six (6) sugar residues. Similarly, a small saponin ion at m/z 767 identified as Stichorrenoside A coelutes with the m/z 1435 ion identified as Variegatuside E. Pseudo-MS3 experiments were performed to gain structural information of the aglycone and was used together with the fragmentation of larger species to deduce the entire structure of intact saponins. Aglycones exist as protonated species in the positive mode and produce significant fragments in MS2 compared to larger saponins which exist as sodiated peaks. For instance, the assignment of m/z 809 parent ion as having the holost-7(8)-ene backbone is supported by the annotation of the resulting product ions.

The majority of the annotated large saponin peaks contain six glycosidic residues with varying combinations of xylose, glucose, and methylglucose. Other known glycosides of *S. cf. horrens* were either not found in the crude extracts or isobutanol fractions, nor do these peaks have intensities high enough to generate fragmentation spectra. A summary of saponins with annotated fragmentation data in the positive and negative mode are shown in Table 2, with at least 2 new saponins that are reported for the first time. The annotation of neutral losses is summarized in S4 Fig. Saponins observed in the positive mode are sodium adducts $[M+Na]^+$, while they exist as formate adducts $[M+CHOO]^-$ in the negative mode. Fragmentation in low-energy CID results in the sequential formation of b- and y-ions based on Domon and Castello's nomenclature of sugar fragments [41]. In the positive mode, labile acetyl groups are observed as neutral loss of acetic acid ($\Delta 60$ Da) which occurs immediately or preceding a monosaccharide loss. The whole sugar chain can also be observed as a sodium adduct through the loss of the whole aglycone. Following this, the loss of individual sugar residues can be

**Table 2. Annotated saponins found in the positive and negative mode.** Fragmentation spectra acquired in the positive and negative mode were annotated through manual fragmentation analysis.

| $[M+Na]^+$ | $[M+CHOO]^-$ | ID | Molecular Formula | Theoretical Mass | ppm diff |
|---|---|---|---|---|---|
| 1433.6407 | 1455.6613 | Unsaturated Variegatuside E | $C_{66}H_{106}O_{32}$ | 1410.6637 | 9.05 |
| 1435.6637 | 1457.6698 | Variegatuside E | $C_{66}H_{108}O_{32}$ | 1412.68237 | 5.97 |
| 1447.6676 | 1469.6797 | 23-acetyl-holost-7(8)-ene; 3-O-[MeGlc-Xyl-Xyl-[MeGlc-Glc]-Xyl] | $C_{67}H_{108}O_{32}$ | 1424.6821 | 3.00 |
| 1475.6589 | 1497.6724 | Stichloroside B2 | $C_{68}H_{108}O_{33}$ | 1452.6748 | 3.90 |
| 1477.6719 | 1499.6859 | Stichloroside B1 | $C_{68}H_{110}O_{33}$ | 1454.69294 | 7.43 |
| 1461.6483 | 1483.6537 | 25,26-didehydro-23-acetyl-7(8)-ene; 3-O-[3-O-Glc-Xyl-Glc-[MeGlc-Glc]-Xyl] | $C_{68}H_{110}O_{32}$ | 1438.6585 | 27.4 |

observed. No peaks can be assigned to the sodiated aglycone in the positive mode and its mass can be calculated as a neutral loss from the parent ion to the ionized sugar chain. Meanwhile, in the negative mode, the loss of acetic acid was not observed. However, there seems to be a mechanism involved wherein the formate ion abstracts a hydrogen molecule from the saponin compound, resulting in a neutral loss of formic acid (Δ46 Da) and the corresponding [M-H]— ion of the saponin. Sequential sugar losses can be observed after until the appearance of the deprotonated aglycone peak. The sequence of neutral losses, intensity of product ions, and the structure of reported saponins in literature were used to deduce the connectivity of the sugar chain [44].

## Discussion

Untargeted metabolomics of *S. cf. horrens* revealed small molecules from different compound classes that play important roles in sea cucumber biology and chemical ecology. Furthermore, the occurrence of these highly bioactive small molecules highlights the potential of developing functional food and therapeutics from *S. cf. horrens* extracts. Phospholipids are ubiquitous constituents as they were observed in both body wall and viscera of samples. These amphiphilic compounds are known to play structural, metabolic, and signaling roles in sea cucumbers and other marine invertebrates [45]. Sea cucumber phosphatidylcholines (PCs) displayed greatest diversity in terms of molecular species, as PCs with one (monoacylated) or two chains (diacylated) of different aliphatic lipid tails are observed. There is a higher relative abundance of single tail phosphatidylcholines as well as free fatty acids in the internal organs. Meanwhile, double tail phosphocholines were more abundant in the body wall. Monoacylated PCs are known to form micellar structures that could aid in the digestion and absorption of nutrients and transport of substances across various organs within the viscera [46, 47]. Meanwhile, double tail lipids form bilayers that are important in maintaining structural integrity of body wall tissues [47].

The fatty acyl make-up of characterized phosphocholines vary in length and degree of unsaturation. Saturated PCs range from chains with n = 13 to n = 22, including many odd numbered acyl substituents. This is in contrast with even-numbered oleic (18:1), linoleic (18:2), and linoleic acid (18:3), which are present in saturated forms. In addition, polyunsaturation was noted for more hydrophobic PCs with n = 20 to n = 26. Analysis in GNPS identified numerous plasmalogens, which correspond to phosphatidylcholines and phosphatidylethanolamines with aliphatic chains attached to sn-2 of the head group through a vinyl linkage [48]. Plasmalogens have antioxidant activity by serving as sacrificial anode-neutralizing reactive oxygen species [48]. Moreover, plasmalogens are also structurally relevant in facilitating membrane fusion [49].

Baseline profiling of these phosphatidylcholines is important as they have been shown to be highly dependent on the geographical location and diet of sea cucumbers [5, 50]. An important food source in aquaculture industries is algae and feeding conditions could be further optimized to increase the yield of desired fatty acids [51, 52]. The polyunsaturated arachidonic acid, eicosapentaenoic acid (EPA), docosahexaenoic acid (DHA), as well as many other omega-3 and omega-6 fatty acids are valued for their cardioprotective and anti-obesity benefits [53]. In addition, phosphocholine-bound PUFAs have greater bioavailability compared to those linked into diglycerides and triglycerides [54]. As a cosmeceutical ingredient, presence of free fatty acids and polyunsaturated PCs are promising since both have been experimentally reported to promote wound healing in vivo [55, 56]. For these reasons, qualitative and quantitative analyses of phosphocholines in sea cucumber extracts and commercially available food and topical products need to be carried out.

More polar than phosphocholines, phosphoethanolamines are associated with higher melting temperatures and thus impart greater membrane fluidity [57]. Furthermore, the cellular role of phosphatidylethanolamines has been established in terms of membrane fusion and participation in cytokinesis during cell division [58]. Aside from these, a recent study implicated phosphatidylethanolamines in the stress response of sea cucumbers [16]. Ding, et al. (2021) described upregulation of Pes in the coelomic fluid of *A. japonicus* prior to gut evisceration [16]. Evidently, higher abundance of phosphoethanolamine substrates significantly altered GPI-anchor biosynthesis and protein assembly leading to the induction of autophagy in *A. japonicus* [16]. Other studies on PE plasmalogens enriched with EPA suggest enhanced cognitive abilities of SAMP8 mice and the suppression of amyloid β-protein formation in CHO-APP/PS1 cells, suggesting other possible applications of PEs extracted from sea cucumbers [59]. Guided by this work, succeeding studies could investigate the role of phosphatidylethanolamines as well as signaling phosphatidylinositols (PI) and phosphatidylserines (PS) on the environmental adaptation of juvenile and adult *S. cf. horrens*.

Aside from phosphatidylethanolamines, another important class of lipids linked to morphological transformation in sea cucumbers are cerebrosides [16]. Determined extensively in both the body wall and internal organs, the biosynthesis of these compounds involving glucosylceramide synthetase were noted to be downregulated during gut evisceration in *A. japonicus* [16]. Accumulation of free ceramides trigger multiple reaction cascades that lead to cell apoptosis [60]. Coupled decrease in the levels of cerebrosides were also related to a decrease in the structural integrity of cell membranes and shrinkage of body walls [53]. Outside their physiological significance, the benefits of sphingolipids on skin have been extensively studied [61]. Duan, et al. (2016), reported the digestion and absorption of dietary cerebrosides into ceramides that facilitate wound healing and improved skin barrier function in mice [62]. Cosmetological assessments in humans also showed enhanced moisture uptake and skin barrier improvement in the presence of ceramides [63]. As a functional food, sea cucumbers rich in cerebroside were noted to improve gut health [62]. Supplementation of Stichopus japonicus extract in mice increased cecal content of immune-enhancing short-chain fatty acids, known to be produced by probiotic microbes [62]. The identified cerebrosides could also very well be the precursors of high molecular weight sphingolipids described in *S. cf. horrens* and related species with neuritogenic and anti-cancer benefits [64–66].

Also annotated in the extracts are compounds containing anionic sulfate groups. The presence of sulfated sterols in *S. cf. horrens* is not a surprising finding since similar compounds have been identified in other sea cucumbers [67, 68], most extensively with the far-eastern *Eupentacta fraudatrix* [68]. These compounds, such as cholesterol sulfate and cholestanol sulfate, are predicted to have chemopreventive and antineoplastic activities, in addition to predicted anti-hypercholesterolemic and wound healing capabilities [69]. *E. fraudatrix* is very similar to *S. cf. horrens* in its ecological response to predators which is in the form of body wall melting [68]. The sulfated compound, beta-sitosteryl sulfate, has been shown to enhance fluidity of bilayers and has huge potential as a skin-softening agent [70]. Meanwhile, the presence of cholesterol and its sulfated analogs could be protective mechanisms of sea cucumbers against cytotoxic saponins it releases. Compared to other triterpenoids, cholesterol and its sulfated version are known to decrease permeability of saponins across cell membranes [71]. The regulation of sterol biosynthesis and corresponding sulfation mechanisms are aspects that could certainly be further studied alongside saponin release or body wall shedding of sea cucumbers in the presence of predatory stress [71].

Using a simple methylene blue colorimetric test, the presence of sulfated compounds was further supported together with their proximate quantification [26]. We observed that sulfated compounds in the viscera are more abundant compared to the body wall. In addition to

sulfated sterols, annotations for additional compounds in the internal organs are sulfated alkanes, also described in *A. japonicus* and *C. frondosa* [72, 73]. Future work could further explore the endogenous role of these metabolites in the viscera or as a signalling kairomone, outside their predicted antimicrobial, antifungal, and antifouling properties [72, 74, 75].

Saponins are primarily used by sea cucumbers for chemical defense against predators, and they have been shown to possess haemolytic and cytotoxic properties [76–78]. In some species, these compounds are concentrated in Cuvierian tubules which are expelled upon predation [79]. *S. cf. horrens* do not possess this structure and thus must rely on other forms of defense. In fact, this species is known to undergo dermal shedding in order to escape predators. Higher concentrations of saponins in the body wall may assist in the escape of the animal after dermal shedding. Moreover, some saponins have been shown to deter predation by acting as pheromones to signal unpalatability [80]. In some cases, the signal may also invite symbionts such as Harlequin crabs to locate the animal [81].

Several studies on saponins from other species show that saponins with hexasaccharide chains are generally more cytotoxic than those with pentasaccharide chains. Moreover, a xylose or quinovose unit as the fifth sugar in the hexoside chain further increases the toxicity of saponins [82]. Along with these, other imaging studies localized the saponins along the body wall of *Holothuria forskali*, suggesting that larger saponins are either concentrated in the epidermis of the animal or are released in the surrounding seawater environment [80]. The efficiency of delivery of these compounds through water can also possibly increase due to its increased hydrophilicity, and thus increase cytotoxicity [82]. The identified saponins, Stichloroside B1, B2 and Variegatuside E, are known to possess potent antifungal activities [77, 78]. In addition, our data also show possibly new sulfated saponin structures. Published bioactivity data of other sulfated saponins also show interesting activities such as antitumor, antiangiogenic, anti-metastatic, and anti-obesity activities, and hence may hint on possible bioactivities of the annotated compounds [83–85]. Other saponin compounds such as Variegatusides and Stichorrenosides with known cytotoxic properties previously found in *S. horrens* in other areas, might also be present in the extracts, however the intensities of some saponin ions are too low to produce sufficient fragmentation data. Thus, further enrichment and purification of saponin compounds are required to obtain an absolute, total, identification of compounds found in the animal. It is also important to note that the metabolites of an animal vary depending on its geographical origin, and hence appropriate metadata should be properly documented when performing comparative analyses [5].

## Conclusion

A rapid characterization workflow of the metabolomic profile of *S. cf. horrens* has been established, providing data that can serve as baseline information for future characterization and comparative studies. Diacyl phosphocholine species were found to be more abundant than monoacyl counterparts, and, together with phosphoethanolamines, may suggest their involvement in the restructuring or remodeling of the dermis during the stress response of the animal. In addition, most of the lipids identified, including polyunsaturated fatty acids, phosphocholines, and cerebrosides are known to have some nutraceutical and medicinal value. In addition to these, triterpene glycosides annotated are known to have antimicrobial and cytotoxic properties and hence warrant further studies. These annotations may be used as a guide for further studies in exploring more specific compounds, or as chemotaxonomic markers especially for phosphocholines and saponins. This broad, general annotation and identification of compounds provides a wider avenue for exploring and furthering the potential of *S. cf. horrens* as a source of high-value biomolecules.

## Supporting information

**S1 Fig. Base peak ion chromatogram of *S. cf. horrens* crude extract.** Metabolite fingerprints were obtained for the body wall and the viscera in the positive (A, B) and negative (C, D) mode respectively.
(TIF)

**S2 Fig. Proposed reaction mechanisms for the annotation of important Mass2Motifs present in dataset from *S. cf. horrens*.** For the reactions, (a) remote H-rearrangement (b) inductive cleavage (c) loss of CH3COOH (d) loss of fatty acyl group (e) loss of ethanolamine combined with elimination of a fatty acyl substituent (f) nucleophilic substitution (g) loss of hexose (h) loss of H2O, formation of conjugated pentadiene carbocation, (i) formation of glucose phosphate anion, (g) dehydration through condensation reaction, (k) homolytic cleavage.
(SVG)

**S3 Fig. Annotation of the product ions obtained from the fragmentation of [M+H+] = 809.4667.** The MS1 and MS2 spectra are shown as (A) and (B) respectively, and the proposed reaction mechanism is shown in (C). Reactions proposed for the fragmentation are as follows: a) loss of the acetyl group ($\Delta$60 Da), b) retro-Diels-Alder at the 2nd ring of the sapogenin, c) opening of the lactone ring followed by a subsequent loss of water ($\Delta$18 Da), and d) decarbonylation ($\Delta$28 Da).
(TIF)

**S4 Fig. MS2 spectra of the saponin ion annotated as Stichloroside B1.** Parent ion in the positive mode is [M+Na]+ = 1477.6719 and the negative mode is [M+CHOO]- = 1499.6859.
(TIF)

**S1 File. Python script calculating cosine score similarities of compounds based on SMILES strings.**
(PY)

**S1 Table. List of putatively identified primary metabolites from *S. cf. horrens*.**
(DOCX)

**S2 Table. List of putatively identified fatty acids from *S. cf. horrens*.**
(DOCX)

**S3 Table. List of putatively identified acylcarnitines from *S. cf. horrens*.**
(DOCX)

**S4 Table. List of putatively identified sphingolipids from *S. cf. horrens*.**
(DOCX)

**S5 Table. List of putatively identified phosphatidylinositols from *S. cf. horrens*.**
(DOCX)

**S6 Table. List of putatively identified phosphatidylserines from *S. cf. horrens*.**
(DOCX)

**S7 Table. List of putatively identified phosphatidylethanolamines from *S. cf. horrens*.**
(DOCX)

**S8 Table. List of putatively identified phosphatidylcholines from *S. cf. horrens*.**
(DOCX)

**S9 Table. Data preprocessing workflow and parameters used for feature-based and ion-identity molecular networking via MZMine.**
(DOCX)

**S10 Table. List of precursor ions common between positive and negative mode analysis.**
(DOCX)

## Acknowledgments

The authors would like to thank the Marine Invertebrate Ecology Laboratory under Dr. Marie Antonette Juinio-Meñez for the collection and maintenance of animals, and the Mass Spectrometry Facility at the Institute of Chemistry, UP Diliman for instrument use.

## Author Contributions

**Conceptualization:** Hiyas A. Junio, Eizadora T. Yu.

**Data curation:** Vicenzo Paolo M. Torreno, Ralph John Emerson J. Molino.

**Formal analysis:** Vicenzo Paolo M. Torreno, Ralph John Emerson J. Molino.

**Funding acquisition:** Eizadora T. Yu.

**Investigation:** Vicenzo Paolo M. Torreno, Ralph John Emerson J. Molino.

**Methodology:** Vicenzo Paolo M. Torreno, Ralph John Emerson J. Molino.

**Project administration:** Eizadora T. Yu.

**Resources:** Hiyas A. Junio, Eizadora T. Yu.

**Supervision:** Hiyas A. Junio, Eizadora T. Yu.

**Visualization:** Vicenzo Paolo M. Torreno, Ralph John Emerson J. Molino.

**Writing – original draft:** Vicenzo Paolo M. Torreno, Ralph John Emerson J. Molino.

**Writing – review & editing:** Vicenzo Paolo M. Torreno, Ralph John Emerson J. Molino, Hiyas A. Junio, Eizadora T. Yu.

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
