## [Decision Letter · Decision Letter 0]

16 May 2023

PONE-D-23-11721Comprehensive metabolomics of Philippine Stichopus cf. horrens reveals diverse classes of valuable small molecules for biomedical applicationsPLOS ONE

Dear Dr. Yu,

Thank you for submitting your manuscript to PLOS ONE. After careful consideration, we feel that it has merit but does not fully meet PLOS ONE’s publication criteria as it currently stands. Therefore, we invite you to submit a revised version of the manuscript that addresses the points raised during the review process.

We look forward to receiving your revised manuscript.

Kind regards,

Anil Bhatia, Ph.D

Academic Editor

PLOS ONE

Journal Requirements:

   "The authors would like to acknowledge the Department of Science and Technology–Philippine Council for 

Agriculture, Aquatic, and Natural Resources Research and Development project entitled “Characterization of High 

Value Biomolecules from the Sea Cucumber Stichopus spp.” under the program “Discovery of High Value 

Biomolecules from the Sea Cucumber Stichopus spp.” for funding; and Dr. Marie Antonette Juinio-Meñez and her 

staff from the Marine Invertebrates Laboratory at the Marine Science Institute, University of the Philippines - Diliman 

for providing samples and engaging in helpful discussions. The UP-Diliman Institute of Chemistry Mass Spectrometry 

Facility where all the LC-MS analyses were performed. DOST PCHRD Drug Discovery of Health Products (DDHP) for 

the purchase of the QTOF instrument."

   "ETY received funding for the study from

the  Department of Science and Technology–Philippine Council for Agriculture, Aquatic, and Natural Resources Research and Development. The funders had no role in study design, data collection, analysis, the decision to publish, or the preparation of the manuscript."

Reviewers' comments:

Reviewer's Responses to Questions

**Comments to the Author**

1. Is the manuscript technically sound, and do the data support the conclusions?

Reviewer #1: Yes

Reviewer #2: Partly

Reviewer #3: Yes

2. Has the statistical analysis been performed appropriately and rigorously? 

Reviewer #1: Yes

Reviewer #2: Yes

Reviewer #3: Yes

3. Have the authors made all data underlying the findings in their manuscript fully available?

Reviewer #1: Yes

Reviewer #2: Yes

Reviewer #3: Yes

4. Is the manuscript presented in an intelligible fashion and written in standard English?

Reviewer #1: Yes

Reviewer #2: Yes

Reviewer #3: Yes

5. Review Comments to the Author

Reviewer #1: This study aimed to investigate the metabolite profile of Stichopus cf. horrens, a valuable sea cucumber species in Southeast Asia with medicinal properties, and identify the compounds responsible for its bioactivities. Through LC-MS/MS experiments and open-access platforms such as GNPS, XCMS, and metaboAnalyst, the study identified various phospholipids, terpenoid glycosides, saponins, sulfated alkanes, and sterols in the crude samples. The study also revealed the abundance and distribution of these compounds in different body parts of the species, suggesting their potential ecological and morphological implications. The presence of terpenoid glycosides and saponins with reported anti-cancer benefits in the body wall was particularly noteworthy. The study provides a useful baseline for further comparative and exploratory studies and highlights the untapped potential of S. cf. horrens as a source of bioactive molecules. This study is novel as it represents the first biochemical analysis of the metabolite profile of S. cf. horrens and provides new insights into the potential bioactive compounds of this species.

The authors must remake figures 2-6. The figures are poor quality and items are not legible.

Reviewer #2: The manuscript is well written but needs revision before it is published.

Reviewer’s comments

The authors in the present manuscript have investigated and characterized the metabolite profile of S. cf. horrens using various experimental and bioinformatics tools. Although the manuscript is well written, the overall motivation is lacking. S. cf. horrens is an important bioactive molecule, but there is not enough literature supporting this. The authors need to revise the manuscript to highlight the significance with some evidence. Is this the first study to characterize the S. cf. horrens?

The figures throughout the manuscript are of very poor quality and it is hard to access the conclusion drawn from the manuscript. The authors decode the presence of various phospholipids in the species, it would be helpful if they can provide the chemical structures as well. Also, the ratio of different types os phospholipids is not mentioned in the manuscript, but this could be a valuable information to access bioactivity and toxicity of the species. Certain abbrevaitions need full forms such as PUFAs, MS2LDA…etc.

Overall the manuscript needs significant revision and improvement before it is published.

Reviewer #3: 1) All the figures, figure labels and figure legends are of low resolution and barely legible.

2) A discussion on how from literature one may assume that the compounds discovered here will have biomedical applications is warranted.

6. PLOS authors have the option to publish the peer review history of their article (what does this mean?). If published, this will include your full peer review and any attached files.

Reviewer #1: No

Reviewer #2: No

Reviewer #3: No

---

## [Author Response · Author response to Decision Letter 0]

23 Jun 2023

Thank you for the insightful comments and suggestions. A major issue in our original submission was the resolution of the figures. We have uploaded high-resolution files of all our figures, however the figures in the compiled PDF are still blurry. We would like to ask the reviewers view the original files via the blue download link in the top right corner of the PDFs. 

Below is our detailed responses to the reviewers

Reviewer #1

This study aimed to investigate the metabolite profile of Stichopus cf. horrens, a valuable sea cucumber species in Southeast Asia with medicinal properties and identify the compounds responsible for its bioactivities. Through LC-MS/MS experiments and open-access platforms such as GNPS, XCMS, and metaboAnalyst, the study identified various phospholipids, terpenoid glycosides, saponins, sulfated alkanes, and sterols in the crude samples. The study also revealed the abundance and distribution of these compounds in different body parts of the species, suggesting their potential ecological and morphological implications. The presence of terpenoid glycosides and saponins with reported anti-cancer benefits in the body wall was particularly noteworthy. The study provides a useful baseline for further comparative and exploratory studies and highlights the untapped potential of S. cf. horrens as a source of bioactive molecules. This study is novel as it represents the first biochemical analysis of the metabolite profile of S. cf. horrens and provides new insights into the potential bioactive compounds of this species. 

The authors must remake figures 2-6. The figures are poor quality and items are not legible.

Response: Figures have been redrawn as vector files and have been scaled accordingly.

Reviewer #2

The authors in the present manuscript have investigated and characterized the metabolite profile of S. cf. horrens using various experimental and bioinformatics tools. Although the manuscript is well written, the overall motivation is lacking. S. cf. horrens is an important bioactive molecule, but there is not enough literature supporting this. The authors need to revise the manuscript to highlight the significance with some evidence. Is this the first study to characterize the S. cf. horrens? 

Response: Stichopus horrens is a particularly interesting organism as it is used as an ingredient for topical salves and exhibits a broad spectrum of body elasticity changes with varying stimuli. Much of the medicinal use of S. horrens in Southeast Asia is in the form of “gamat” oil or gels used primarily for wound healing. Although there is literature showing that S. horrens extracts facilitate wound healing, very little known about what specific compounds are in the sea cucumber extracts. To our knowledge, this is the first study to perform a comprehensive structural characterization of metabolites extracted from S. cf. horrens. We have added this information in the introduction (Page 3, lines 9-10). We also added references to other biochemical studies on similar or related compounds (Page 20 lines 12-15; Page 21 lines 14- 17; Page 23 lines 5-9) as additional evidence on their potential biomedical use.

2. The figures throughout the manuscript are of very poor quality and it is hard to access the conclusion drawn from the manuscript. 

Response: The figures have been redrawn and scaled accordingly as vector images. 

3. The authors decode the presence of various phospholipids in the species, it would be helpful if they can provide the chemical structures as well. 

Response: The compound names we listed in the supplemental tables are conventional shorthand nomenclature of lipids. We have added footnotes in the tables as a guide to deducing structures from the shorthand notations. 

4. Also, the ratio of different types os phospholipids is not mentioned in the manuscript, but this could be a valuable information to access bioactivity and toxicity of the species.

Response: The focus of this study is the baseline characterization of the S. horrens metabolome. While we agree that it is valuable information, quantitation was not part of this study. However, we plan to quantify select PLs in the future. 

5. Certain abbreviations need full forms such as PUFAs, MS2LDA...etc. 

Response: We have revised the manuscript to define the abbreviations listed, except for MS2LDA, which derives its name from the latent Dirichlet allocation algorithm but was not directly stated in the reference.

6. Overall the manuscript needs significant revision and improvement before it is published.

Reviewer #3 

All the figures, figure labels and figure legends are of low resolution and barely legible

Response: The figures have been redrawn and scaled accordingly as vector images.

A discussion on how from literature one may assume that the compounds discovered here will have biomedical applications is warranted.

Response: We have added references to other biochemical studies on the same or related compounds (Page 20 lines 12-15; Page 21 lines 14-17; Page 23 lines 5-9) as additional evidence on their potential biomedical use.

---

## [Decision Letter · Decision Letter 1]

14 Aug 2023

PONE-D-23-11721R1Comprehensive metabolomics of Philippine Stichopus cf. horrens reveals diverse classes of valuable small molecules for biomedical applicationsPLOS ONE

Dear Dr. Yu, 

Thank you for submitting your manuscript to PLOS ONE. After careful consideration, we feel that it has merit but does not fully meet PLOS ONE’s publication criteria as it currently stands. Therefore, we invite you to submit a revised version of the manuscript that addresses the points raised during the review process.

We look forward to receiving your revised manuscript.

Kind regards,

Anil Bhatia, Ph.D

Academic Editor

PLOS ONE

Journal Requirements:

Reviewers' comments:

Reviewer's Responses to Questions

**Comments to the Author**

1. If the authors have adequately addressed your comments raised in a previous round of review and you feel that this manuscript is now acceptable for publication, you may indicate that here to bypass the “Comments to the Author” section, enter your conflict of interest statement in the “Confidential to Editor” section, and submit your "Accept" recommendation.

Reviewer #2: All comments have been addressed

Reviewer #3: (No Response)

Reviewer #4: (No Response)

2. Is the manuscript technically sound, and do the data support the conclusions?

Reviewer #2: Yes

Reviewer #3: Yes

Reviewer #4: Partly

3. Has the statistical analysis been performed appropriately and rigorously? 

Reviewer #2: Yes

Reviewer #3: Yes

Reviewer #4: No

4. Have the authors made all data underlying the findings in their manuscript fully available?

Reviewer #2: Yes

Reviewer #3: Yes

Reviewer #4: Yes

5. Is the manuscript presented in an intelligible fashion and written in standard English?

Reviewer #2: Yes

Reviewer #3: Yes

Reviewer #4: Yes

6. Review Comments to the Author

Reviewer #2: The comments have been addressed in a satisfactory manner. I would suggest that the upcoming study that the authors plan of doing should focus on the quantitative analysis of the PLs as well.

Reviewer #3: The font sizes and resolution of the following figures are required to be improved to make them legible:

Figure 2 axes labels

Fig 3A-numbers, Figure 3B and D labels in the pie chart and name of metabolites

Fig 4. labels in the network diagrams are not legible. Please improve the resolution of the figure.

Figure 6 Chemical reaction schemes and chemical structures. Similar figures in the SI have very good resolution.

I recommend accepting the manuscript only after the above revisions.

Reviewer #4: 10. The manuscript has many other grammatical and formatting errors which are to be fixed. Manuscript needs proper revision before submission. Statistical significance of data is lacking. Figures are not clear.

7. PLOS authors have the option to publish the peer review history of their article (what does this mean?). If published, this will include your full peer review and any attached files.

Reviewer #2: No

Reviewer #3: No

Reviewer #4: **Yes: **Pallavi Agarwal

---

## [Author Response · Author response to Decision Letter 1]

4 Oct 2023

Reviewer #2

The comments have been addressed in a satisfactory manner. I would suggest that the upcoming study that the authors plan of doing should focus on the quantitative analysis of the PLs as well. 

Response:

Quantitative analysis is planned for future projects. Thank you.

Reviewer #3

1. The font sizes and resolution of the following figures are required to be improved to make them legible:

Figure 2 axes labels

Fig 3A-numbers, Figure 3B and D labels in the pie chart and name of metabolites.

Fig 4. labels in the network diagrams are not legible. Please improve the resolution of the figure.

Figure 6 Chemical reaction schemes and chemical structures. Similar figures in the SI have very good resolution.

I recommend accepting the manuscript only after the above revisions.

Response:

There seems to be a problem in compiling the PDF for review. We were assured by PLOSONE Editorial Assistants that this is normal. We request the reviewers to donwnload the high-resolution figures through the blue link found in the upper right hand corner of the figure pages. THese are high resolution figure source filesand have passed the figure resolution requirement of PLOS One. Nevertheless, we have corrected the figures and increased the font sizes according to the comments.

Reviewer # 4

The manuscript has many other grammatical and formatting errors which are to be fixed. Manuscript needs proper revision before submission. 

Statistical significance of data is lacking. Figures are not clear. 

Response:

It would be helpful if the reviewer points out example pages and lines that pertain to said errors. Nevertheless, we have reviewed the manuscript and found no significant grammatical nor formatting errors. 

As for the requirement for statistical significance of the data presented, we would like to highlight that the focus of the paper is the identification of metabolites from S. horrens.. The data analysis programs we used ( e.g.,GNPS) have built-in metrics for statistical significance and confidence of mass assignments. Further, the goal of the study is to annotate a baseline metabolome of the species, which may serve as reference for future comparative and quantitative analyses. 

Figures have been addressed in accordance with the comments of the Reviewers from the first round of review. There seems to be a problem in compiling the PDF for review. The figures we uploaded and available through the blue link ( found in the upper right hand corner of the figure pages) are high resolution and have passed the resolution requirement of PLOS One. Nevertheless, we have corrected the figures and increased the font sizes according to the comments. We would like to humbly request the reviewers to download the files through the blue link provided to properly scale resolution, should the figure in the pdf still come out as blurry.

Reviewer Comment 

1. The present study reveals the metabolite profile of S.cf. horrens using bioinformatics tools. The manuscript satisfies the current title but the overall reflection of importance of molecules for lacking. Addition of more literature regarding the relevance of the study will be better. It requires more clear view.

Response:Thank you for this suggestion. We did include the relevance of our study in the introduction section ( page 3 lines 6-9, 12-14) and in the discussion section ( page 18 lines 12-14, page 19 lines 22-23, page 20 lines 2-3, 11-17, page 21 lines 5-13, 17-20, page 23 lines 8-12). But briefly, S. horrens is a high-value sea cucumber commodity in Asia and this species is the source of “gamat”, the wound-healing medicinal remedy. Unfortunately, even given its widespread use, there is no comprehensive metabolome reported for this species and possible attribution for the bioactivity of the sea cucumber extract. This study represents a baseline analysis of this sea cucumber metabolome. As for the molecules we identified, we have included literature references to support the bioactivity of the reported molecules

2. In Materials and methods page 5, under heading Sampling and extraction of secondary metabolites from S.cf. horrens; age of ten S.cf.horrens is not mentioned and also, whether sampling was done in triplicate or not.

Response: We did not indicate the age of the samples as we used broodstock animals for the study. We revise the methodology to include that the animals were mature adults. The metabolome we report is from a total of ten animals collected from the same hatchery.

3. In Materials and method page 7, under heading Bioinformatics analysis; Data was pre-processed using MZmine v2.52 should be Data was pre-processed using MZmine 2.52 similarly for Cytoscape too.

Response :We have revised the manuscript to read MZmine 2.52 and Cytoscape 3.8.2. Thanks.

4. In Materials and method page 8, under heading Quantification of sulfated compounds; model number for the Shimadzu Multi-scanner needs to be reported.

Reponse:Thank you for this. We have revised the manuscript to reflect the model and make of the multiscanner.

5. All the figures are of low quality. unpublishable quality is lacking. No conclusion can be drawn from the figures. The figure legends need to be rewritten. The legends should be succinct.

Response: Please see the explanation ( previous page). These have been addressed accordingly.

6. Figure 3 lacks the color scale which indicate which color corresponds to what. Although it is written in legend, but figures need to be self-explanatory.

Response: These have been addressed accordingly.

7. Page 14, legend for figure 6 is to be removed. Page 16 again has legend for Fig6.

Response:These have been corrected accordingly.

8. All the chemical formulas in Page 14-16 needs to have numerals as subscript.

Response:These have been corrected accordingly.

9. Statistical significance of data is lacking in all figure legends.

Response:This has been addressed earlier. Please see our remarks on Reviewer 4 comments found in previous pages.

10. The manuscript has many other grammatical and formatting errors which are to be fixed. Manuscript needs proper revision before submission.

Response: This has been addressed earlier, please see remarks on comments on reviewer 4 comments in the previous page.

---

## [Decision Letter · Decision Letter 2]

3 Nov 2023

Comprehensive metabolomics of Philippine Stichopus cf. horrens reveals diverse classes of valuable small molecules for biomedical applications

PONE-D-23-11721R2

Dear Dr. Yu,

We’re pleased to inform you that your manuscript has been judged scientifically suitable for publication and will be formally accepted for publication once it meets all outstanding technical requirements.

Kind regards,

Anil Bhatia, Ph.D

Academic Editor

PLOS ONE

Additional Editor Comments (optional):

Reviewers' comments:

Reviewer's Responses to Questions

**Comments to the Author**

1. If the authors have adequately addressed your comments raised in a previous round of review and you feel that this manuscript is now acceptable for publication, you may indicate that here to bypass the “Comments to the Author” section, enter your conflict of interest statement in the “Confidential to Editor” section, and submit your "Accept" recommendation.

Reviewer #3: All comments have been addressed

Reviewer #4: All comments have been addressed

2. Is the manuscript technically sound, and do the data support the conclusions?

Reviewer #3: Yes

Reviewer #4: Yes

3. Has the statistical analysis been performed appropriately and rigorously? 

Reviewer #3: N/A

Reviewer #4: Yes

4. Have the authors made all data underlying the findings in their manuscript fully available?

Reviewer #3: Yes

Reviewer #4: Yes

5. Is the manuscript presented in an intelligible fashion and written in standard English?

Reviewer #3: Yes

Reviewer #4: Yes

6. Review Comments to the Author

Reviewer #3: (No Response)

Reviewer #4: I would recommend to write which statistical tools (algorithm) were used by data analysis programs. Statistical analysis is a must for quantitative analysis. But, even for baseline metabolome it needs to be mentioned in manuscript if though done by data analysis programs. It helps in reproducibility of results.

7. PLOS authors have the option to publish the peer review history of their article (what does this mean?). If published, this will include your full peer review and any attached files.

Reviewer #3: No

Reviewer #4: No

---

## [Editor Report · Acceptance letter]

10 Nov 2023

PONE-D-23-11721R2 

Comprehensive metabolomics of Philippine *Stichopus cf. horrens* reveals diverse classes of valuable small molecules for biomedical applications 

Dear Dr. Yu:

I'm pleased to inform you that your manuscript has been deemed suitable for publication in PLOS ONE. Congratulations! Your manuscript is now with our production department. 

Kind regards, 

on behalf of

Dr. Anil Bhatia 

Academic Editor

PLOS ONE